The postcranial skeleton of Boreogomphodon (Cynodontia: Traversodontidae) from the Upper Triassic of North Carolina, USA and the comparison with other traversodontids

Liu Jun liujun@ivpp.ac.cn 1 2
Schneider Vincent P. 3
Olsen Paul E. 4
1 Key Laboratory of Vertebrate Evolution and Human Origins of Chinese Academy of Sciences, Institute of Vertebrate Paleontology and Paleoanthropology, Chinese Academy of Sciences , Beijing , China
2 University of Chinese Academy of Sciences , Beijing , China
3 North Carolina Museum of Natural Sciences , Raleigh , NC , United States of America
4 Lamont-Doherty Earth Observatory, Columbia University , Palisades , NY , United States of America
Marsicano Claudia
Electronic publication date: 2017 Sep 14
Publication date: 2017
Volume: 5
Electronic Location ID: e3521
Received 2016 Dec 18; Accepted 2017 Jun 8
Copyright: ©2017 Liu et al.
Copyright year: 2017
Copyright holder: Liu et al.
License: This is an open access article distributed under the terms of the Creative Commons Attribution License, which permits unrestricted use, distribution, reproduction and adaptation in any medium and for any purpose provided that it is properly attributed. For attribution, the original author(s), title, publication source (PeerJ) and either DOI or URL of the article must be cited.
License URL: https://creativecommons.org/licenses/by/4.0/

Keywords: Postcranial skeleton, Cynodontia, Boreogomphodon, Traversodontidae

Funding: Chinese Academy of Sciences XDPB 0501 Columbia University IRG-China/South Africa Research Cooperation Programme CS08-L02/95449 This study was supported by Chinese Academy of Sciences (XDPB 0501), Columbia University through a Faculty Fellowship, and IRG-China/South Africa Research Cooperation Programme (CS08-L02/95449). The funders had no role in study design, data collection and analysis, decision to publish, or preparation of the manuscript.

==============================
Postcranial remains of Boreogomphodon from the Upper Triassic of North Carolina are described and compared to those of other known traversodontid cynodonts. The postcranial skeleton of Boreogomphodon is characterized by four sacral ribs, simple ribs lacking costal plates, the extension of the scapular neck below the acromion process, a short scapular facet on the procoracoid, a concave anterior margin of the procoracoid, humerus entepicondyle with smooth corner, and the presence of a fifth distal carpal. Four types of ribs are identified among traversodontids: ‘normal’ form, tubercular rib, costal plate, and Y-shaped rib. Fossorial behavior is suggested for traversodontids with elaborate costal plates. Within Traversodontidae, the procoracoid is relatively small; the anterior process of the iliac blade extends anteroventrally to different degrees in different taxa, which facilitates retraction of the femur; and the limb bones show allometric growth in terms of length and width.

Introduction

Cynodontia is a diverse clade and represents an important therapsid radiation that is especially noteworthy for including crown-group mammals. During the Early Triassic, eucynodonts diverged into two clades, Cynognathia and Probainognathia. The former includes many taxa with buccolingually (transversely) expanded postcanine teeth whereas the latter, a clade mostly represented by sectorial-toothed members, gave rise to mammaliaforms by the Late Triassic (Hopson & Kitching, 2001; Liu & Olsen, 2010). The most successful Triassic lineage of cynodonts, in terms of their species richness and specimen abundance, is Traversodontidae (Abdala & Ribeiro, 2010; Liu & Abdala, 2014). Traversodontids are characterized by postcanine teeth whose crowns are labiolingually expanded and ellipsoid-to-rectangular in outline. The upper postcanines bear a deep occlusal basin. The lower postcanines are more quadrangular than the upper ones in outline and usually contain an anteriorly positioned transverse crest (Liu & Abdala, 2014).

Boreogomphodon jeffersoni was originally described based on a left maxilla with postcanine teeth from the Tomahawk Creek Member of the Vinita Formation (Carnian) in the Richmond Basin of the Newark Supergroup, Chesterfield County, Virginia (Sues & Olsen, 1990). The same locality subsequently produced numerous cranial elements and a small number of postcranial bones that were also referred to this species (Sues & Hopson, 2010; Sues & Olsen, 1990). Plinthogomphodon herpetairus was named based on the cranial remains of a small cynodont preserved as gut content in the partial skeleton of the archosaur Postosuchus alisonae from the “Lithofacies Association II” (Upper Triassic: Norian) of the Deep River basin of the Newark Supergroup in North Carolina (Sues & Hopson, 2010; Sues, Olsen & Carter, 1999). Later, Liu & Sues (2010) suggested that Plinthogomphodon might prove to be a subjective junior synonym of Boreogomphodon, although there are slight differences in the structure of the lower gomphodont postcanines. Additional traversodontid remains, including cranial and well-preserved postcranial elements, from the Pekin Formation (upper Carnian or lower Norian) of Merry Oaks Quarry, Triangle Brick Company, Chatham County, North Carolina were tentatively referred to Boreogomphodon jeffersoni (Liu & Sues, 2010). However, a detailed study of the skull and mandible is necessary before a more secure taxonomic identification can be made.

Previous therapsid research has focused on the skull, only a few postcranial characters have been analyzed phylogenetically, and those characters remain uncoded for most species (Huttenlocker, Sidor & Angielczyk, 2015; Kammerer, Fröbisch & Angielczyk, 2013; Liu & Olsen, 2010). The relative neglect of the postcranial skeleton also accurately describes the history of traversodontid studies (Liu & Abdala, 2014), although this has recently begun to change. Postcranial descriptions are published for the following traversodontid taxa: Exaeretodon argentinus (Bonaparte, 1963), Pascualgnathus polanskii (Bonaparte, 1966), Massetognathus pascuali (Jenkins Jr, 1970b), Luangwa drysdalli (Kemp, 1980), Menadon besairiei (Kammerer et al., 2008), Andescynodon mendozensis (Liu & Powell, 2009), Protuberum cabralense (Reichel, Schultz & Soares, 2009), and Massetognathus ochagaviae (Pavanatto et al., 2015). Here we describe the postcranial skeleton of Boreogomphodon from the Pekin Formation of North Carolina and use it to review postcranial variation across Traversodontidae.

Material NCSM 20698, skull with lower jaws, most of the postcranial skeleton; NCSM 20711, skull with lower jaws, anterior part of the postcranial skeleton including ∼27 nearly continuous vertebrae; NCSM 21370, skull with lower jaws, and partial postcranial skeleton including nearly complete left forelimb.

Description

Axial skeleton

NCSM 20698 includes a nearly complete vertebral column, with 27 mostly articulated vertebrae. The total number of presacral vertebrae is probably 24, and there are four sacrals.

Cervical series

It is difficult to distinguish between cervical and dorsal vertebrae in non-mammaliaform cynodonts due to the presence of cervical ribs. Brink (1954) differentiated cervicals from dorsals based on the presence of the tallest neural spine on the first thoracic vertebra and identified five cervicals in Thrinaxodon. Jenkins Jr (1971) distinguished between cervical and dorsal vertebrae based on the orientation of the zygapophyses and transverse processes and the morphology of the neural spine. He used these criteria to conclude that, like crown mammals, Thrinaxodon and Cynognathus contained seven cervicals (Jenkins Jr, 1971). Neither set of criteria can be applied to the material under study here and thus the number of cervicals remains uncertain.

Figure 1 Boreogomphodon (NCSM 20711), cervical vertebrae and ribs 2 to 5 in lateral view.

Abbreviations: 2–5, cervical 2–5; d, dentary; cl, clavicle; cr, cervical rib; ns, neural spine; pp, posterior process. Scale bar equals 1 cm.

The cervical vertebrae are hidden in NCSM 20698 but exposed in NCSM 20711, so the following description is based on the latter specimen (Fig. 1). No proatlas or atlas can be identified. The centra of the second (axis), third and fourth cervicals are broken, with only the right sides partially preserved. The fifth centrum shows only the anterior end. However, their neural spines are well-preserved. The axis centrum is nearly seven millimeters long; two millimeters longer than that of any successive vertebra. The axial neural spine is a broad blade, approximately 10 mm long. Its strongly concave dorsal margin agrees with Menadon besairiei (Kammerer et al., 2008) but differs from that taxon in having a convex rather than concave posterior half. Transversely, the spine is thin through the middle portion, but its thickness increases anteriorly and posteriorly, ending in a tuberosity. Menadon and now Boreogomphodon are the only traversodontids whose axial neural spine has been described.

The neural spine of the third cervical (C3) is canted posteriorly behind the posterior process of the axial spine, with a height equal to the posterior margin of the axial neural spine. Thus, it is proportionately taller than that of Menadon besairiei (Kammerer et al., 2008). On the fourth through seventh cervicals, the neural spines are tall, narrow, and slightly canted posteriorly. The neural spines on the third through fifth cervicals abruptly taper toward the apex and are triangular in lateral view. The neural spines on the sixth and seventh vertebrae are distinctly taller than those that precede them. The transverse processes of the third and fourth cervicals are stout and directed posterolaterally and ventrally.

The second (axis) through fifth cervical ribs are preserved in NCSM 20711. In lateral view, each rib is a short curved rod that is directed posteroventrally. Each rib is approximately 7 mm in length; slightly longer than the corresponding centra. Vertebral articulations of the ribs are not exposed.

Dorsal series

Based on the structure of the posterior ribs, the dorsal vertebral column in traversodontid cynodonts is either relatively undifferentiated (e.g., Exaeretodon sp., Bonaparte, 1963) or divided into a “thoracic” and “lumbar” region (e.g., Pascualgnathus polanskii, Bonaparte, 1966). The ribs are poorly preserved in known specimens of Boreogomphodon, therefore assessment of any division in the dorsal column is impossible.

In NCSM 20698, 14 dorsal vertebrae are exposed and form the basis for the following description (Figs. 2 and 3). The centrum is amphicoelous, approximately circular in cross-section, and slightly constricted at mid-length. Its ventral surface is smooth without a keel. There are no intercentra. The anteroposterior lengths of the centra of the anterior dorsal vertebrae measure approximately 5 mm and slightly increase posteriorly, reaching 6 mm for the more posterior dorsal vertebrae except for last three, where they are 5.3 mm.

Figure 2 Boreogomphodon (NCSM 20698), anterior dorsal vertebrae and ribs.

In (A) ventral and (B) ventrolateral views, the anterior is right; posterior dorsal vertebrae in (C) dorsal, (D) left lateral and (E) ventral views, the anterior is left. Two parts form a continue series. Scale bars equal 1 cm.

Figure 3 Boreogomphodon (NCSM 20698), posterior dorsal, sacral and anterior caudal vertebrae and ribs.

In (A) ventrolateral and (B) ventral views; the left ischium in lateral view. Anterior is right. Abbreviations: dr, dorsal rib; poz, postzygapophysis; prez, prezygapophysis; sr, sacral rib; sv, sacral vertebra. Scale bar equals 5 mm.

In lateral view, the neural arch joins the centrum along an irregular suture. The pedicles are incised anteriorly and posteriorly to form vertebral notches, of which the latter are invariably more deeply incised. No anapophyses are present. The transverse processes are reduced into small bulges on the pedicles. Their positions vary along the dorsal column. Anteriorly, the processes arise from the anterior half of the pedicles, close to the prezygapophyses, whereas on the last three dorsals, they arise from the pedicles at a point adjacent to the postzygapophyses.

The articular facets of the prezygapophyses mainly face medially and slightly dorsally against the ventrally and laterally directed articular facets of the postzygapophyses. The prezygapophyses are thin blades that extend slightly beyond the level of the anterior margin of the centrum. The postzygapophyses extend posteriorly from the base of the neural spine beyond the posterior margin of the centrum. The neural spines decrease remarkably in height posteriorly. They shift to the posterior ends of the neural arches, extendposteriorly beyond the posterior rim of the centrum, and lie above the prezygapophyses of the succeeding vertebra. The morphology of the posterior dorsal vertebrae is generally characterized by a nearly flat dorsal surface not including the neural spine (Figs. 2C and 2D).

The dorsal ribs articulate with costal foveae on the anterior dorsal vertebrae in Massetognathus and Menadon (Jenkins Jr, 1970b; Kammerer et al., 2008), which are intervertebral in position. The situation in Boreogomphodon is not clear-cut. The ribs lack any structural specialization. The length of anterior dorsal ribs is approximately 33 mm in NCSM 20711, whereas the length of an isolated rib is approximately 20 mm in NCSM 21370 (both specimens have similar skull length).

Sacral series

The lack of a preserved ilium somewhat complicates identification of the sacral series. In Thrinaxodon and Cynognathus, the sacral centra are similar in length to those of the lumbar region, but tend to be narrower and more constricted at the middle (Jenkins Jr, 1971). Following this criterion, four sacral vertebrae are identified in NCSM 20698 (Fig. 3). The transverse processes and the zygapophyses of the first sacral vertebra are more slender than those of the last dorsal vertebra. The zygapophyseal facets are nearly parallel to the parasagittal plane.

The first left sacral rib and second right sacral rib are still articulated with the centra. The first sacral rib is wider than the last dorsal rib, whereas the latter appears to be wider than the second sacral rib. The second right sacral rib is approximately 6 mm long and has a distinctly expanded distal end. One isolated element (sr3? in Fig. 3) is identified as a sacral rib with an expanded distal end. It is 7 mm long, 4 mm wide proximally, and 6 mm wide distally. Posteriorly, there is another sacral rib (sr4?), which is more slender than the anterior ones. As in the sacral ribs of other cynodonts, the capitulum and tuberculum are confluent and the sacral ribs are not fused to the corresponding vertebrae.

Caudal series

Three anterior caudal vertebrae are exposed in ventral view in NCSM 20698 (Fig. 3). Each centrum is approximately 4 mm long.

Pectoral girdle

Most elements of the pectoral girdle are preserved in NCSM 20711 (Fig. 4A), including the left clavicle, left scapula, interclavicle, and coracoids. The left procoracoid and coracoid are firmly connected along a serrated suture, whereas the incomplete right procoracoid is isolated. Most elements are also preserved in NCSM 20698 (Figs. 4B–4E), but only the left scapula and right procoracoid are well exposed.

Figure 4 Boreogomphodon, shoulder girdles.

(A) NCSM 20711 mainly in ventral view. NCSM 20698, left scapula in (B) lateral, (C) posterior, and (D) medial views; (E) right procoracoid in lateral view. Abbreviations: acr, acromion; ar, anterior ridge; bi, fossa for origin of biceps muscle; C, clavicle; cb, fossa for origin of coracobrachialis muscle; cc, concavity for articulation with medial end of clavicle; COR, coracoid; d-t min, insertion for deltoideus plus teres minor muscle complex; f prc, procoracoid foramen; H, humerus; IC, interclavicle; lr, lateral ridge; pr, posterior ridge; PRC, procoracoid; prc (cor), procoracoid articular surface for coracoid; prc (s), procoracoid articular surface for scapula; S, scapula; spc, fossa for supracoracoideus muscle. Scale bars equal 5 mm.

Scapula

The scapula is relatively small. In NCSM 20698 (Figs. 4B–4D), it is 20 mm tall, compared to a humerus length of 29 mm. The scapula is bowed laterally, with an elongate blade whose lateral surface is marked by a narrow but deep fossa. This fossa extends from the dorsal part of the blade to approximately its midpoint. The fossa served as the origin for the deltoid and teres minor muscles, as reconstructed by Kemp (1980) and Kemp & Parringon (1980).

The posterior border of the scapula extends close to the edge of the glenoid as a clearly defined crest, although at the base it is merely a low ridge and not a free flange as along the anterior border. The anterior flange extends only for about two thirds of the dorsal portion of the scapular blade, ending above the scapular neck. The dorsal portion of this flange is a thin sheet of free-standing bone. The acromion process extends in a position similar to that of Luangwa or Menadon but is less developed (Kammerer et al., 2008; Kemp, 1980). The scapula is constricted and elongate between the acromion process and the glenoid portion, and the neck is more pronounced than in Massetognathus (Jenkins Jr, 1970b) and Exaeretodon (Bonaparte, 1963; Jenkins Jr, 1970b).

The base of the scapula bears a slightly convex semicircular glenoid facet. The articular surface is rough, indicating an extensive cartilaginous covering in life. It faces posterolaterally as well as ventrally.

Procoracoid

The procoracoid is identified by the presence of a procoracoid foramen. A bone in NCSM 20698 is identified as a right procoracoid in lateral view (Fig. 4E). It differs from the procoracoid of other known traversodontids in its possession of an acute anterior tuberosity.

The bone is an ax-shaped plate (Figs. 4A and 4E). The procoracoid foramen is close to the concave anterodorsal border of the bone. The articular surface for the scapula forms an obtuse angle to the anterodorsal border. The dorsal edge is short, and the procoracoid does not participate in the formation of the glenoid.

Anterior to the foramen is a shallow fossa that accepted at least a portion of the supracoracoid muscle. An anteriorly directed ridge, which is more prominent in NCSM 20698, separates the supracoracoid origin from the remainder of the lateral surface of the procoracoid. Ventral to this ridge, a crescentic depression faces anteroventrally and probably represents the origin of the biceps brachii muscle. The ventral margin of this fossa forms a sharp, strongly convex keel. The procoracoid protrudes anteriorly far beyond the procoracoid-scapula contact, forming a swollen terminal tuberosity. This protrusion increases the area for the attachment of biceps and possible coracobrachialis.

Coracoid

The coracoid contacts the procoracoid in NCSM 20711 (Fig. 4A). It is larger than the procoracoid. Although not preserved in articulation with the scapula, its robust anterodorsal margin would have contacted the supraglenoid buttress above and formed the coracoid portion of the glenoid. The posterodorsal margin of the coracoid is concave. The posterior end forms a slightly elongated process, which is incomplete but probably terminated in a tubercle for the origin of the coracoid head of the triceps. The ventral side of the lateral surface of the coracoid is indented to form a shallow fossa for the origin of the coracobrachialis muscle. The fossa extends onto the posteroventral corner of the lateral surface of the procoracoid.

Clavicle

The lateral half of the clavicle is a slender rod that is directed dorsolaterally. The medial half consists of a gradually expanding, spatulate plate, which is directed medially and horizontally (Fig. 4A). The long axes of the medial and lateral portions intersect at an angle of about 150°. The medial plate is bordered by rather sharply defined edges. The posterior edge becomes distinct from the clavicular shaft at approximately the midpoint of the clavicular shaft where the shaft has its greatest curvature. The anterior edge is set off from the clavicular shaft more abruptly, giving the medial plate a slightly asymmetrical appearance. The medial plate of the left clavicle is articulated with the anteroventral concavity of the interclavicle. The clavicular facet for the acromion on the distal end is not well exposed but it contacts the left scapula.

Interclavicle

The interclavicle is similar to that of Thrinaxodon (Jenkins Jr, 1971). It is cruciform with a long posterior ramus and a short transverse bar (Fig. 4A). The anterior triangular part is slightly convex, and the anterior and lateral ridges are not so distinct. The concavities defined by the anterior and lateral ridges are shallow, and the left one is in contact with the medial end of the clavicle. The posterior rectangular portion of the interclavicle is nearly flat except for a low but distinct posterior ridge in the center. The posterior margin is slightly expanded transversely.

Forelimb

The forelimb is preserved in NSCM 20698, 20711 and 21370, including two articulated hands. An articulated manus is rare among traversodontids being known previously only in Exaeretodon (Bonaparte, 1963). The bones of NCSM 21370 are better ossified than those in the other two specimens. NCSM 21370 includes a nearly complete left hand, in which a set of nine carpals and most of the phalanges are preserved. The following description is mainly based on this specimen.

Humerus

The humerus is highly similar to that in most traversodontid cynodonts except Exaeretodon (Bonaparte, 1963) (Fig. 5). The width of its proximal end, measured from the lesser tuberosity to the region of the greater tuberosity, equals approximately one third the total length of the humerus. The maximum width across the epicondyles is about 45% of humerus length (Table 1); this ratio is exceeds 50% in Exaeretodon (Bonaparte, 1963).

Figure 5 Boreogomphodon, humeri.

Right humerus (NCSM 20698) in (A) ventral, (B) anterior (lateral), (C) dorsal, and (D), posterior (medial) views; distal part of left humerus (NCSM 21370) in (E) ventral, (F) distal, and (G) dorsal views. Abbreviations: bi gr, bicipital groove; cp, capitulum; d c, deltopectoral crest; ec, ectepicondyle; en, entepicondyle; f en, entepicondylar foramen; g t, greater tuberosity; h, humeral head; l t, lesser tuberosity; sc f, supracondylar flange; th, trochlea. Scale bar equals 1 cm.

Table 1 Measurements of humeri of traversodontids (in mm) and their ratios.

Taxa	Specimen	Length	PW	DW	S1	S2	PW/L	DW/L	
Boreogomphodon	NCSM 20698	28.5	9	12.4	3.5	3.1	0.32	0.44	
	NCSM 20711	32.6		15.5				0.48	
	NCSM 21370	35	11.5	16.5			0.33	0.47	
Andescynodon	PVL 3894	36.5	12	18.5	5.5	4.7	0.33	0.51	
	PVL 3890	37	13.5	18.5	6	5	0.36	0.5	
	PVL 4426	45.6	15	20	6.7	5.5	0.33	0.44	
Massetognathus	PVL 4613	47	13	19		5	0.28	0.4	
	PVL 5444	55	16	20	6.5	6.5	0.29	0.36	
	UNIPAMPA 0625	61	24	30	8	9	0.39	0.49	
Pascualgnathus	65-vi-18-1	56.5	21	23	8.5	6.5	0.37	0.41	
Luangwa	OUMNH TSK-121	94	37	36	12	11	0.39	0.38	
Exaeretodon	UFRGS no number	116	44	55	22	16	0.38	0.47	
	61-VIII-2-6a	165	70		29	27	0.42		
	61-VIII-2-16	182	75	113		35	0.41	0.62	
	PVL 2467	184	82	91	35	32	0.45	0.49	
	PVL 2554	186	82	100	37	29	0.44	0.54	
Notes.

PW proximal width

DW distal width

S1 shaft minimum width in dorsoventral direction

S2 shaft minimum width in anteroposterior direction

The proximal half of the humerus is composed of two planes, the deltopectoral crest and the shaft, which intersect along the broad bicipital groove at an angle of around 105°. The short middle shaft connecting the expanded proximal and distal ends is triangular in cross-section. The expanded distal half of the humerus is triangular in dorsal view. The proximal and distal articular ends of the humerus are as well-ossified as in large-sized cynodonts, differing from those of the similar-sized Thrinaxodon (Jenkins Jr, 1970b; Jenkins Jr, 1971; Kemp & Parringon, 1980). The rounded humeral head is at the center of the strap-shaped surface of the proximal end. Its boundary is not obvious because the articular surface is confluent with the lesser tuberosity medially and with the proximal margin of the deltopectoral flange laterally. The greater tuberosity is hard to discern. The lesser tuberosity is set apart from the head by a slight depression across the strap-shaped proximal articular surface. The broad deltopectoral flange amounts to nearly half the total length of the humerus. It is thin and flat, but thickens towards the junction with the middle shaft. The free margin of the flange curves distinctly ventrally. The dorsal bony ridge on the dorsal side extending across the flange as in ?Cynognathus is evident, although it is not clearly preserved (Jenkins Jr, 1971).

Arising from the robust ectepicondylar region, a thin supracondylar flange extends proximally as well as somewhat dorsally in NCSM 20698. Its anterior margin is straight, not curved. The ectepicondylar foramen does not open on the dorsal surface, but a concave fossa appears to be present on the ventral surface of the proximal side of the flange, in particular, on the right humerus of NCSM 20711. It indicates that the ectepicondylar foramen is closed. The long, oval entepicondylar foramen is enclosed by a stout bar of bone, which arises from the entepicondylar region and continues to the deltopectoral flange (Fig. 5E).

The capitulum is bulbous and contributes to the thickness of the ectepicondylar region. Its articular surface is entirely confined to the ventral aspect of the humerus where its surface is confluent with those of the trochlea distomedially and ectepicondyle laterally. A bulbous ulnar condyle lies between the capitulum and the entepicondyle, which is only slightly smaller than the capitulum. A shallow, narrow groove represents the trochlea. The dorosoventral principal axis of this groove is slightly oblique as in Massetognathus (Jenkins Jr, 1970b). The ulnar condyle is well developed and contacted the sigmoid notch of the ulna. The thickness of the ectepicondylar region is much greater than that of entepicondylar region. The entepicondyle is a stout process but less dorsally developed and thus continuous with the posteromedial margin, as in Exaeretodon, contrasting with the angular shaped entepicondyle of Luangwa and Pascualgnathus (Bonaparte, 1963; Bonaparte, 1966; Kemp, 1980).

Figure 6 Boreogomphodon, forelimbs.

Left forelimb (NCSM 21370), (A) manus in plantar view, ulna and radius in posterior view; (B) carpus and metacarpus in plantar view; (C) ulna and radius in anterior view; (D) carpus in dorsal view. Right forelimb (NCSM 20698), (E) manus in plantar view, ulna and radius in posterior view; (F) a few digitals in dorsal view, it continues with digits in (E). Abbreviations: c, centrale; dc, distal carpal; i, intermedium; mc, metacarpal; paf, proximal articular facet; pc, proximal centrale; R, radius; U, ulna; r, radiale; u, ulnare. Digits in Roman numbers. Scale bars equal 5 mm.

Radius

The radii are articulated with the ulnae in all specimens (Fig. 6). The radius is a sigmoid bone with expanded proximal and distal ends. The distal half of the shaft is curved posteriorly and slightly medially to facilitate its crossing over the anterior aspect of the ulna. The proximal articular facet is oval or nearly semicircular, with a nearly straight edge along the posteromedial side. The facet forms a shallow concavity sloping medially. On the posterolateral aspect of the proximal end a protuberance bears a facet for articulation with the ulna. A flange or ridge for insertion of the biceps brachii is not evident on the radii of NCSM 20698 and this region is not exposed in NCSM 21370. The distal end of the radius is triangular in outline, expanding gradually toward the distal articular facet. Along the anterolateral side of the rim is a tuberosity for contact with the distal end of the ulna.

Ulna

The ulna is a sigmoid bone with an anteroposteriorly expanded proximal end (Figs. 6C and 6E). In lateral view, its shaft is narrow, with the distal end evenly expanded mediolaterally and the proximal end expanded primarily anteriorly. An olecranon process is not developed. As preserved, the semilunar notch is a relatively shallow, slightly concave facet with a rather straight posterior margin and a nearly semicircular anterior margin. This facet is inclined mainly medially.

The ulnar flange on the medial side of the shaft for the interosseous ligament (Jenkins Jr, 1970b) is not well exposed. The radial notch is represented by a fossa on the medial side of the anterior surface, immediately distal to the sigmoid facet. The posterior surface of the ulna is smooth.

Carpus

Nine carpals have been identified, including the ulnare, intermedium, radiale, two centralia, and four distal carpalia; at least one distal carpal is missing. The ulnare is a stout bone, longer than wide in anterior and posterior view. It is constricted between its proximal and distal ends and the medial edge is longer than the lateral edge in plantar view. Its proximal end articulates not only with the distal end of the ulna but also the intermedium. The thickness of the bone is greater dorsally than ventrally so that the bone is nearly triangular in lateral view. The distal surface bears a small facet for articulation with the fourth and fifth distal carpals; the medial surface bears a deep groove for the reception of the lateral centrale (c2). The intermedium is a small rounded bone and only exposed in dorsal view (Figs. 6D and 6E). It underlies the ulna and lies between the radius and the ulnare. With its cartilaginous component, it would have likely contacted the lateral centrale and possible radiale distally. The radiale is stout, with an irregular quadrangular shape, and is best exposed in ventral view (Figs. 6B–6E). Its proximal surface is a rounded facet for articulation with the radius. It contacts two centralia with anteromedial (dorsomedial) and anterolateral (dorsolateral) facets; distally it touches the distal carpal 2 (Figs. 6B, 6D and 6E). This was likely the original relationship in life because the same pattern is observed in both NCSM 20698 and 21370. No pisiform is present. The medial centrale (c1) is rectangular, with its proximodistal length shorter than those of the other axes. The lateral centrale (c2) is a flat, nearly square bone. The medial and part of its ventral surfaces are covered by the radiale, so the lateral centrale is exposed as a small triangle in ventral view (Fig. 6B). Its distal end articulates with distal carpals 3 and 4. Although only distal carpals 2 to 4 are preserved, there were probably five in total because distal carpal 1 is present in all described traversodontid manus (Bonaparte, 1963; Jenkins Jr, 1971; Kemp & Parringon, 1980). All distal carpalia are somewhat nodular. The third and fourth distal carpals have the same size, and both are slightly larger than the second and much larger than the fifth carpal.

Metacarpal

Four metacarpals are preserved in NCSM 21370. All five metacarpals are preserved in NCSM 20698 but the fifth is incomplete. The metacarpals are elongate and dumbbell-shaped and vary only in shaft length, with IV>III>II>I. In ventral view, the metacarpals appear nearly symmetrical, their proximal ends flaring somewhat less laterally than the distal ends. The proximal articular facet of each metacarpal is gently convex, whereas the distal facet is flat.

Phalanges

In NCSM 21370, the fourth and fifth digits have three phalanges, whereas in NCSM 20698, the first digit has two phalanges, the second digit has at least two phalanges, and the third digit has three phalanges (Figs. 6A, 6E and 6F). The inferred digital formula of the manus is 2-3-3-3-3. The phalanges are more slender than those of Exaeretodon and Cynognathus (Jenkins Jr, 1971) and flat in lateral view. Proximal phalanges are elongate and dumbbell-shaped with the articular ends similar in size and proportions. They are moderately constricted at mid-length. The penultimate phalanges are also elongate and dumbbell-shaped. The proximal and distal ends are similar, with proximal end being slightly wider. The midlength constriction is much narrower than on the proximal phalanges. The articular surface for the ungual phalanx is concave. The fourth and fifth ungual phalanges are slender, tapering cones (Fig. 6A). The proximal articular facet is convex in ventral view. The first and third ungual phalanges are short (Fig. 6F).

Pelvic girdle

The left ischium in NCSM 20698 is the only element of the pelvic girdle that could be studied (Figs. 3 and 7). The ischium is composed of a proximal head and a ventromedially enlarged plate. Its acetabular surface is oval, concave, and occupies the anterolateral surface of the head. The articular facets for the ilium and pubis are convex. The ischium is slightly constricted below the head, forming a short neck with the plate. The ischial plate is fan-shaped, with an expanded distal part. The dorsal margin of the plate is mediolaterally expanded by a ridge extending from the middle of the lateral acetabular rim to the posterodorsal corner of the plate. The dorsal surface is smooth without an obvious groove. The portion of the plate below the ridge is thin. The posterior edge of the plate is short and straight in lateral view. Anteroventrally, there is the long ischial symphysis. The anterior edge of the ischium is smoothly concave, forming the posterior border of the obturator foramen. The anteroventral corner has no evidence of contact with the pubic plate; this suggests the place is not completely ossified.

Figure 7 Boreogomphodon (NCSM 20698), ischium and hindlimb.

(A) hindlimb; (B) left ischium in lateral view; (C) left femur in ventral view, left tibia and left fibula in posterior views; (D) right pes in plantar view. Abbreviation: as, astragalus; ca, calcaneum; cu, cuboid; ecc, ectocuneiform; enc, entocuneiform; FE, femur; FI, fibula; if, intertrochanter fossa; mec, mesocuneiform; mit, minor trochanter; mjt, major trochanter; mt1∼mt4, metatarsal 1∼4; n, navicular; p2, second phalange; st, sustentaculum tali; tc, tuber calcis; TI, tibia. Digits in Roman numbers. Scale bars equal 1 cm.

Hindlimb

The hindlimb is known in NCSM 20698. It includes the incomplete left femur, the left tibia, the proximal half of the left fibula, the nearly complete right fibula, and the articulated right pes. An articulated pes has only been reported in Exaeretodon (Bonaparte, 1963) and NHMUK R9391, possibly Scalenodon (see discussion) (Jenkins Jr, 1971).

Femur

The femur is exposed in ventral and anteromedial views (Figs. 7A and 7C). It has a moderately slender shaft and expanded articular end. The femur is straight for most of its length but has a strong dorsomedially angle proximally. Due to this proximal dorsal bowing of the proximal end of the shaft, the head, which is bulbous and almost hemispherical, is reflected medially. The head bears rough texture typical of bone supporting a cartilaginous cap. There is a crest connecting the head with the major trochanter, resulting in a semicircular outline for the proximal end of the femur (Fig. 7C).

A deep intertrochanteric fossa lies on the ventral surface between the head and major trochanter and represents the point of insertion for the pubo-ischio-femoralis externus muscle. Distal to the fossa, the minor trochanter runs distally along the ventral side of the shaft. It is a prominent flange that extends for about 6 mm and gradually merges into the bone at about mid-shaft. Distal to the minor trochanter, the anterior and the ventral surfaces of the femur are separated along an angular intersection. In cross-section, the shaft is nearly oval at mid-length; its thickness from the extensor to flexor surface is about 3.2 mm and its transverse width is 4.8 mm.

Tibia

The left tibia is articulated with the proximal half of the fibula (Figs. 7A and 7C). It is almost only exposed in posterior view. The shaft of the tibia is flat and bowed medially. The proximal and distal ends are expanded mainly laterally so that the lateral margin is concave and the medial margin is slightly convex. Due to the poor ossification, the facets on the proximal articular end are not clearly defined. The lateral margin of this end is thickened and protuberant. The distal end terminates in a convex oval facet set at a right angle to the long axis of the shaft (Fig. 7C). The bone is 22 mm in length, and its width is slightly more than 4 mm proximally and distally.

Fibula

The fibula has a slender shaft with expanded ends and is bowed laterally (Figs. 7A and 7C). The proximal articular end is poorly ossified. The shaft is narrow proximally but gradually expands anteroposteriorly distally. The distal articular surface is oval in outline and convex. It contacts the concave articular surface formed by the calcaneum and astragalus.

Tarsus

The shape, number, and proportion of the tarsal elements (Fig. 7D) are similar to those in an unidentified cynodont from the Manda beds of Tanzania (NHMUK R9391) (Jenkins Jr, 1971).

The calcaneum is distoproximally elongate, but in contrast to NHMUK R9391, its distal head is slightly narrower than the proximal tuber calcis. A separate element seems to be present between the calcaneum and the astragalus. Based on the comparison with NHMUK R9391, it is identified as a process of the calcaneum. This stout process is about half of the width of the calcaneum and covers the astragalus ventrally. The sustentaculum tali lies dorsal and distal to the proximal facet for the astragalus, and a distinct calcaneal sulcus separates them. The calcaneum is constricted distally to form an articular surface exclusively for the cuboid.

The exact shape of the astragalus is unknown because it is covered by the calcaneum. It looks like a bean in ventral view. Its anterior edge is concave with a distal end that articulates with the navicular. The medial edge is slightly convex dorsomedially.

The navicular (centrale) is an irregular oval element. Its plantar surface is nearly flat or slightly convex. It articulates proximally with the astragalus and distally with the first, second, and third distal tarsalia (ento-, meso-, and ectocuneiforms) and probably with the fourth distal tarsal (cuboid) as well.

There are four distal tarsalia—entocuneiform, mesocuneiform, ectocuneiform, and cuboid (from medial to lateral). The entocuneiform is nearly rectangular but its distal side is slightly wider than the proximal side. It articulates distally with metatarsal I and laterally (apparently) with metatarsal II. The mesocuneiform, the smallest of the tarsalia, is wedge-shaped. Distally, it articulates with metatarsal II and proximally it has a short contact with the navicular. The ectocuneiform is triangular in outline and articulates distally with metatarsal III. The cuboid is smaller than the entocuneiform and similar to the navicular in size and shape. Distally it articulates with metatarsal IV and possibly V, laterally with the ectocuneiform and possibly the navicular, and proximally with the calcaneum.

Metatarsal

Four metatarsals are present but the distal end of the metatarsal I is missing (Fig. 7D). Metatarsals II, III, and IV are nearly similar in size and shape. Their distal ends are wider than the proximal ends and the bones have slightly constricted shafts. The distoproximal lengths are: II, 5.4 mm; III, 6.2 mm; IV, 6.0 mm.

Phalanges

Only a few phalanges are preserved, so the digital formula is unknown (Fig. 7D). In contrast with NHMUK R R9391 (Jenkins Jr, 1971), the proximal phalanges are long and dumbbell-shaped with a median constriction. The lengths of these elements are close to those of the metatarsals.

Comparison and Discussion

The traversodontids are a diverse group including forms with skulls that range from a few centimeters in length to more than 40 cm (Huttenlocker, 2014; Liu, 2007). Their postcanine teeth are highly variable in shape, which is the source for most of the diagnostic characters (Liu & Abdala, 2014). The postcranial elements also show considerable variation among traversodontids. Here we summarize the postcranial features from previous studies and personal observation.

Axial skeleton

Vertebral column

The number of vertebrae is poorly known in traversodontids due to poor preservation and/or insufficient preparation. The number of presacral vertebrae is 28 in Exaeretodon argentinus and E. riograndensis (Bonaparte, 1963; Oliveira, Schultz & Soares, 2007), possibly 26 (>23) in Massetognathus pascuali (Jenkins Jr, 1970b), at least 16 in Protuberum cabralense (Reichel, Schultz & Soares, 2009), and ∼24 (>20) in Boreogomphodon. Among presacral vertebrae, seven cervicals are identified in the former two species, as well as in Thrinaxodon, Cynognathus (Jenkins Jr, 1971), and most extant mammals. The cervical vertebrae of traversodontids, as well as in all non-mammaliaform cynodonts with known cervical series (e.g., Kemp & Parringon, 1980; Oliveira, Soares & Schultz, 2010), have cervical ribs. In mammals, the dorsal vertebral column is divided into a thoracic and lumbar series based on the conservation of articulated ribs in the former. Two series are also recognized in some cynodonts based on rib morphology (Jenkins Jr, 1970b; Jenkins Jr, 1971). Because the rib morphology of the dorsal series varies among traversodontids, no common criterion is applicable. The sacral vertebrae are defined on the basis of their rib contact with the medial surface of the iliac blade. The number of sacral vertebrae varies from six or seven in Exaeretodon (Bonaparte, 1963; Oliveira, Schultz & Soares, 2007), three or four in Pascualgnathus (Bonaparte, 1966) (Fig. 8C), six in Massetognathus pascuali, although the last ribs do not directly link to the iliac plate (Jenkins Jr, 1970b), four in Andescynodon (Liu & Powell, 2009), and possibly four in Boreogomphodon. Based on the current view of phylogenetic relationships (Liu & Abdala, 2014), the common ancestor of traversodontids should have four sacral vertebrae. According to the presacral vertebral count in known traversodontids, it is inferred that the presence of more than 20 presacrals is characteristic of traversodontids.

Figure 8 Different types of ribs in Traversodontidae.

(A) type I, Exaeretodon argentinus (PVL2554); (B) type II, Protuberm cabralensis (UFRGS PV1010T); (C) type III, Pascualgnathus polanskii (MLP 65-VI-18-2); (D) type IV, Masetognathus pascuali (PVL 5443); (E) Scalenodon angustifrons (NHMUK R9391). Scale bars equal 1 cm.

Ribs

Plesiomorphically simple ribs are generally retained in early synapsids, with complex ribs appearing in cynodonts. There is a diversity of rib morphology within the traversodontid cynodonts, with different types sometimes coexisting in the same individual.

The dorsal ribs in Traversodontidae can be divided into four basic types (Fig. 8). A Type I or ‘normal’ rib is one in which the shaft is slender with a slightly expanded proximal end (Fig. 8A). All ribs of Exaeretodon, Boreogomphodon and the anterior dorsal ribs of Massetognathus belong to this type. At least some Type I ribs are likely present in all traversodontid species. A Type II, tubercular rib, exhibits protuberances on its dorsal border (Fig. 8B). Type II is documented only in Protuberum (Reichel, Schultz & Soares, 2009). A Type III rib is modified as costal plate. These ribs conform to a complex morphological gradient wherein the anteroposterior width of the plate and the shaft distal to the plate are variable in the same individual (Fig. 8C). This type is present in Andescynodon, Pascualgnathus, Luangwa, Menadon, Traversodon, and Protuberum (Barberena, 1981; Bonaparte, 1966; Kammerer et al., 2008; Kemp, 1980; Liu & Powell, 2009; Reichel, Schultz & Soares, 2009). It is also present in the basal cynodonts Thrinaxodon and the basal Cynognathia, including Cynognathus, Diademodon, and trirachodontids. The presence of this rib type is considered plesiomorphic in traversodontids (Crompton & Parrington, 1955; Jenkins Jr, 1971). Compared to Thrinaxodon and trirachodontids (NMQR3521), the distal end of the costal plate does not form a double-layered recurved surface. A Type IV rib is bifurcate with a Y-shaped distal end (Fig. 8D). Type IV is only known at the posterior end of the dorsal series in Massetognathus (Jenkins Jr, 1970b; Pavanatto et al., 2015).

Anterior dorsal ribs of traversodontids do not bear costal plates as in Cynognathus, Diademodon, and triracodontids, whereas the posterior dorsal ribs are represented by costal plates in most genera, except Massetognathus, Exaeretodon, and Boreogomphodon. The posterior dorsal ribs are generally shorter than the anterior dorsal ribs, but the transformation in length is smooth in most species and no clear differentiation on thoracic and lumbar region can be made possible other than in Massetognathus. Jenkins Jr (1971, p 55) identified the first lumbar vertebra in Thrinaxodon on the basis of the loss of a rib shaft distal to the costal plate. In traversodontids with type III ribs, this criterion can be applied (e.g., Kemp, 1980).

Generally, sacral ribs are similar in length and have a distal expansion to connect with the iliac blade. However, in Massetognathus pascuali, the first sacral rib (Jenkins Jr, 1970b: fig. 2A, S1) is similar in shape to the last lumbar, and the last sacral rib (Jenkins Jr, 1970b: fig. 2A, S6) is too short to contact the iliac blade (Jenkins Jr, 1970b). The first sacral rib has a more expanded distal end than subsequent ones. The caudal ribs are synostosed to the vertebrae and their shafts direct posterolaterally.

NHMUK R9391 from the Manda Formation is associated with bones of the probainognathian Aleodon and of a few traversodontid species. This specimen only features Type III ribs (Fig. 8E). These ribs are essentially similar to the posterior dorsal ribs of Andescynodon and Pascualgnathus. Because this form of rib is unknown in Probainognathia, it is considered here to be diagnostic of traversodontids.

Jenkins Jr (1971) reviewed the epaxial muscles in reptiles and mammals, associating the costal plates in cynodonts with a well-developed iliocostalis muscle. He suggested two functions for them. The first is related to locomotion. The coastal plates would have provided larger insertional area for attachment of the muscle; and assisted the lateral flexure of the vertebral column. The second function would be the provision of intrinsic strength to the vertebral column by the imbrication of successive ribs. He connected this function with the reinforcement of lumbar region of mammals, which promote the ability of transmit thrust force. Kemp (1980) analyzed function of the costal ribs in Luangwa. He suggested that Luangwa has no lateral movement of the vertebral column because vertebral column is effectively rigid in this plane. He proposed two advantages: the first one is maintenance of the momentum; the second one is the improvement on the maneuverability. However, the curled skeletons of Thrinaxodon such as BP/1/2776 indicate the presence of a considerable lateral movement of the vertebral column even if it has costal plates.

In mammals, xenarthrous vertebrae are perhaps an adaptation for fossorial behavior (Gaudin & Biewener, 1992). Expanded ribs may increase the stability of the vertebral column, and are a common character in fossorial mammals (Jenkins Jr, 1970a). Groenewald, Welman & MacEachern (2001) showed that Trirachodon excavated burrows, and Damiani et al. (2003) demostrated that Thrinaxodon inhabited burrows too. Trirachodon and Thrinaxodon have both costal plates and anapophyses. The anapophyses are associated to Type III ribs in all known traversodontids; this perhaps suggests a fossorial behavior for these species. In Massetognathus, the zygapophyseal facets on posterior dorsals are oriented at angles of around 45° but the anapophyses are absent. So the bending both in lateral and dorsoventral directions is permitted. The posterior process on posterior dorsal ribs maintains the tendency of reinforcement of the lumbar region but reduced to a lighter structure to acquire higher mobility. Although the ribs of Protuberum are special in the presence of dorsal tubercles, the basic pattern is the same as that of other traversodontids with Type I ribs. In Protuberum, the posterior dorsal ribs have larger costal plates than other traversodontids; they overlap each other to form a connected plate. This is the most rigid vertebral column in the group and must have provided increased protection for the internal organs. The surface tubercles perhaps indicate defensive structures. On the other hand, the similar sized Exaeretodon adopted another strategy as their ribs are reduced to normal costal type I. Perhaps only Boreogomphodon has a truly lumbar region in all known traversodontids. The lumbar vertebrae, and possibly all dorsal vertebrae, can rotate in sagittal and horizontal planes, indicating that the vertebral column was able to bend laterally and dorsoventrally. The lumbar vertebrae are more massive than thoracic vertebrae.

Shoulder girdle

Interclavicle and clavicle

The interclavicle and the clavicle are known only in Exaeretodon argentinus (Figs. 9A and 9B; Bonaparte, 1963: fig. 16), Massetognathus pascuali (Figs. 9C and 9D; Jenkins Jr, 1970b: fig. 5), and Boreogomphodon (Fig. 4); the clavicle is also reported in Andescynodon (Liu & Powell, 2009) and Pascualgnathus (Bonaparte, 1966). As in Thrinaxodon (Jenkins Jr, 1971), the interclavicle of Boreogomphodon and Massetognathus is cruciate with an elongate posterior ramus. In Exaeretodon, it is laterally expanded with a short posterior ramus, so the width is similar to the length. No notable difference in the clavicle has been observed between traversodontid species.

Figure 9 Interclavicles and clavicles.

Exaeretodon argentinus (PVL 2467): (A) right clavicle in ventral view; (B) interclavicle and left clavicle in dorsal view; Massetognathus pascuali (PVL 4613): interclavicle and clavicles in (C) dorsal and (D) ventral views. Scale bars equal 1 cm.

Scapulocoracoids

Other than Boreogomphodon, the scapulocoracoids were reported in Andescynodon (Liu & Powell, 2009), Exaeretodon (Bonaparte, 1963), Luangwa (Kemp, 1980), Massetognathus (Jenkins Jr, 1970b; Pavanatto et al., 2015), Menadon (Kammerer et al., 2008), Pascualgnathus (Bonaparte, 1966), and Traversodon (Von Huene, 1936–1942) (Fig. 10).

Figure 10 Scapulocoracoids of various cynodonts.

Cynognathus (NHMUK 2571) in (A) lateral and (B) medial views; Diademodon (UMZ T 502) in (C) lateral and (D) medial views; Pascualgnathus (MLP 65-VI–18-1) in (E) lateral view; Boreogomphodon (NCSM 20698) in (F) lateral view; Andescynodon mendozensis (PVL 4428) in (G) lateral view; Exaeretodon argentinus (PVL 2554) in (H) lateral view; Luangwa (OUMNH TSK 121) in (I) lateral and (J) medial views; Massetognathus pascuali (PVL 4613) in (K) lateral and (L) medial views; Menadon besairiei (FMNH PR 2444) in (M) lateral and (N) medial views; Traversodon (GPIT RE 1069) in (O) lateral and (P) medial views. Scale bars equal 1 cm. (M, N, O, P image credit: Christian F. Kammerer).

In Cynognathus and Diademodon, the scapula is not constricted below the acromion process and the anteroposteriorly shorter portion lies above the acromion process (Jenkins Jr, 1971: fig. 17; Figs. 10A–10D). In Traversodontidae, the scapular blade is constricted below the acromion process, forming an anteroposteriorly short neck (Fig. 10) that provides extra space for the insertion of the supracoracoideus muscle. Kemp (1980) described the acromion process of Luangwa as more reflected laterally than that of Diademodon-Cynognathus (Figs. 10A, 10C and 10I). This condition is represented in all traversodontids with well-preserved scapula. The acromion process is reconstructed very high in the scapula of Pascualgnathus (Fig. 10E; Bonaparte, 1966). However, that portion of the bone is poorly preserved in the specimen and here is interpreted as part of the scapular flange based on personal observation.

The procoracoid participates into the glenoid in Luangwa and Pascualgnathus (Figs. 10E and 10J); it reaches but does not participate in the glenoid in Massetognathus, Menadon, and perhaps Andescynodon (Figs. 10G and 10K–10N); and it is far from the glenoid in Exaeretodon, Boreogomphodon, and Traversodon (Figs. 10F, 10H, 10O, and 10P).

The shape of the procoracoid is variable within this group. The ventral margin of procoracoid is confluent with that of coracoid, forming a convex flange in Andescynodon, Massetognathus, Menadon, possibly in Boreogomphodon and Pascualgnathus; while it is roughly straight in Luangwa. The anterior margin of the procoracoid is convex in Luangwa, and Menadon as in Cynognathus and Diademodon, nearly straight or slight concave in Andescynodon, Boreogomphodon, and perhaps Exaeretodon and Massetognathus.

The procoracoid foramen is close to the articular surface with the scapula in this group, whereas it lies in the anterior corner of the bone, far from the articular surface with the scapula in Cynognathus (Figs. 10A and 10B). It is closer to the articular surface with the coracoid than the anterior margin in Luangwa, Massetognathus (contra Jenkins Jr, 1970b: fig. 6), and Menadon whereas it is closer to the anterior margin of the bone in Exaeretodon (PVL 2554) and Boreogomphodon.

The coracoid is irregularly quadrilateral or approximately triangular in outline. Its posterior process ends in a tuberosity. The tuberosity is short and mainly ventral to the glenoid in Luangwa, Traversodon, and possibly Exaeretodon; but long and distinctly posterior to the glenoid in Andescynodon, Pascualgnathus, Massetognathus, and Menadon. The dorsomedial margin of the coracoid is shorter than that of the procoracoid in Luangwa and possibly Pascualgnathus, nearly equal to that of Massetognathus and Menadon, and is longer than that of Boreogomphodon and perhaps in Andescynodon.

Forelimb

Humerus

The humeri are preserved in many species. The basic shape is the same in this group as in Cynognathus or Diademodon (Fig. 11). The proximal half of the humerus is roughly triangular in most species, whereas it is roughly trapezoid in Exaeretodon for the development of a flange on the posteromedial surface. The articular surface is confluent medially with the lesser tuberosity and laterally with the proximal margin of the deltopectoral flange. The lesser tuberosity is better developed in Luangwa than in other species. The deltopectoral crest reflects laterally in different degrees but this structure is easily deformed during fossilization, and it is uncertain how much of the observed difference is due to deformation.

Figure 11 Traversodontid left humeri in ventral view.

(A) Andescynodon mendozensis (PVL 3894); (B) Pascualgnathus polanskii (MLP65-VI-18-1); (C) Luangwa drysdalli (OUMNH TSK121); (D) Boreogomphodon (NCSM 20698); (E) Massetognathus pascuali (PVL 4241); (F) Exaeretodon argentinus (PVL 2467). (B, D, E) are reflected as left side. Scale bars equal 1 cm.

The relative width of the humerus varies with the length (Table 1). The basic functions of the bones are the support of the body against gravity and to attach the muscles. The diameter (width) of a supporting bone as the humerus should increase with the increase of length (Christiansen, 1999); this is shown by the positive allometry of the proximal width (PW), the distal width (DW), and the sum of the shaft minimum width in anteroposterior and dorsoventral directions (S1 + S2) relative to the length (L) (Fig. 12). The scaling is close to 1.2 other than the one related to the distal width. The regression function for L (Y) to S1 + S2 (X) is: Y = 5.7695X0.833(R2 = 0.984). In the mammalian humerus, the scaling for the least circumference to the length is 0.76 for all mammals, 0.83 for small mammals under least squares regression (Christiansen, 1999). Here, the sum of S1 and S2 can be used as a lineal approximation of the least circumference, so this scaling (0.83) can be compared with that of small mammals. This scaling (l∞d0.83) is intermediate between geometric similarity (isometry: l∞d) and elastic similarity (l∞d0.67), far from stress similarity (l∞d0.5) (Christiansen, 1999). It shows that the humeral growth strategy is similar to that of small mammals. The point of Luangwa appears to be an outlier (Fig. 12), indicating that its shaft is slenderer than the normal humerus.

Figure 12 Regression of various width to humeral length (L).

(A) the proximal width (PW), (B) the distal width (DW), (C) the sum of humeral shaft minimum width in dorsoventral and anteroposterior directions (S1 + S2).

Ulna and radius

As the humerus, the ulna and the radius are robust in large specimen and slender in small specimens. The inter-specific difference is distinct on the proximal side of the ulna (Fig. 13). The ossified olecranon process is absent in all but Exaeretodon, in which the relative length of the olecranon is about fifteen percent of remaining portion of the ulna (MACN 18063, PVL 2467) (Bonaparte, 1963). The length of the ulna is about 68% of the humerus in two specimens of Boreogomphodon, the ratio of ulna/humerus in Exaeretodon, Massetognathus, or Pascualgnathus is greater than 76% (Table 2). If the olecranon portion is excluded, the ratio in Exaeretodon (PVL2467) is similar to that of Boreogomphodon.

Figure 13 Right ulnae of traversodontids in lateral view.

(A) Boreogomphodon (NCSM 20698); (B) Andescynodon (PVL 3890); (C) Pascualgnathus polanskii (MLP 65-VI-18-1); (D) Massetognathus pascuali (PVL 5444); (E) Exaeretodon argentinus (PVL 2467). (A, E) are reflected as right side. All scale bars equal to 1 cm.

Table 2 Measurements of humeri and ulnae of traversodontids (in mm) and their ratios.

Taxon	specimen	Humerus	Ulna	Ratio	
Exaeretodon	PVL 2467	184	146	0.79	
Exaeretodon	PVL 2554	186	144	0.77	
Massetognathus	PVL 5444	55	44	0.80	
Pascualgnathus	MLP 65-VI-18-1	56.5	43	0.76	
Boreogomphodon	NCSM 20698	28	19	0.68	
Boreogomphodon	NCSM 21370	35	24	0.69	

Manus

The manus is only known in Boreogomphodon and Exaeretodon (Bonaparte, 1963: fig. 20). Most carpals are identified in both species, except for the fifth distal carpal in Exaeretodon, and the pisiform in Boreogomphodon. The pisiform is a large element in Thrinaxodon and Diademodon (Jenkins Jr, 1971, p 127), but smaller in Exaeretodon. Even if this bone was ossified in Boreogomphodon, it would have been too small to be observed. Besides Boreogomphodon, a separate fifth distal carpal is only known in one specimen of Thrinaxodon among non mammaliaform cynodonts (Jenkins Jr, 1971; Parrington, 1933). When present, the fifth is the smallest of the distal carpals. It is lost or fused in Exaeretodon. The digital formula of Exaeretodon is 2-3-3-3-3; whereas the preserved elements in Boreogomphodon are 2-1-3-3-3. Because Cynognathus, Diademodon, and Cricodon also have a digital formula of 2-3-3-3-3, it is safe to infer this formula is conserved among all cynognathians, including Boreogomphodon (Crompton & Parrington, 1955).

Pelvis

The pelvis was described in Andescynodon (Liu & Powell, 2009), Exaeretodon (Bonaparte, 1963), Luangwa (Kemp, 1980), Massetognathus (Jenkins Jr, 1970b; Pavanatto et al., 2015), Menadon (Kammerer et al., 2008), Pascualgnathus (Bonaparte, 1966), and NHMUK R9391 (Jenkins Jr, 1971) (Fig. 14).

Figure 14 Pelvises in lateral view.

(A) Scalenodon angustifrons (NHMUK R 9391 = TR 8); (B) Pascualgnathus polanskii (MLP 65-VI-18-1); (C) Andescynodon mendozensis (PVL 3894); (D) Luangwa drysdalli (OUMNH TSK121); (E) Menadon besairiei (FMNH PR 2444); (F) Massetognathus pascuali (PVL 4442); (G, H) Exaeretodon argentinus (PVL2554). (A, H) are reflected from right side. (I) a trirachodontid (NMQR 3521) right ilium. All scale bars equal to 1 cm.

The ilium in traversodontids is clearly different from that of Cynognathus or Diademodon (Jenkins Jr, 1971). The dorsal (vertebral) margin is nearly straight or slightly concave rather than convex in all well-preserved ilia. In a trirachodontid (NMQR 3521), this part looks still convex (Fig. 14I). The angle between the ventral margin of the anterior and posterior processes of the iliac blade is around 120° in NHMUK R9391 and Pascualgnathus, about 150° in Andescynodon, Luangwa, Massetognathus, and Menadon, nearly 180° in Exaeretodon; so the neck between the blade and the base is narrower and more obvious in NHMUK R3521 and Pascualgnathus. In Cynognathus and Diademodon, the neck is wide and short as Massetognathus; but in a trirachodontid (NMQR 3521), the neck is narrow and pronounced as in Pascualgnathus (Fig. 14H). In traversodontids, the ventral margin of the anterior process of the blade is nearly parallel to the dorsal margin. The anterior part of the blade is narrowly rounded and somewhat spoon-shaped, with the exception of Exaeretodon where the anterior part is widely rounded and ax-shaped (Fig. 14G). The anterior process is short in NHMUK R9391 (less than the diameter of the acetabulum), relatively long in Pascualgnathus (between 1 to 1.5 times of the diameter of the acetabulum), and long in other species (greater than 1.5 times of the diameter of the acetabulum). The posterior process is long in all known species other than Exaeretodon, where its length is less than the diameter of the acetabulum.

The morphological features of the ilium and the rib of NHMUK R9391 show this specimen representing a traversodontid species with primitive characters. Two genera, Scalenodon and Mandagomphodon, have been referred to Traversodontidae based on materials from the Manda Formation (Crompton, 1972; Hopson, 2014; Liu & Abdala, 2014). Within their named species, S. angustifrons is far more basal than other species. If we accept the correlation of skull and postcranial features, this specimen could be referred to S. angustifrons.

The lateral surface of the blade is concave, forming a fossa, which lies mainly on the anterior process. The anterior process in Luangwa and Massetognathus features a laterally reflected ventral margin (Kemp, 1980; J Liu, pers. obs., 2006: PVL 4442) that enhance the fossa on the anterior process. With the shape of iliac blade of Exaeretodon, the center of the fossa is close to the anterior margin of the blade. We interpret this fossa as the origin for ilio-femoralis (gluteal) muscle. Jenkins Jr (1971) did not observe muscle markings on the lateral surface of iliac blade in Cynognathus, and he suggested that the origin of the ilio-femoralis (gluteal) muscle was in the fossa anterodorsal to the acetabulum. Kemp (1980) disagreed, suggesting this muscle occupied most of the lateral surface of the iliac blade in Luangwa. With the extension of the anterior process, and the anterior position of the fossa as in Exaeretodon, the muscle is disposed more horizontally and enjoys a greater volume, which results in an increased retraction force on the femur.

One problem with the ilium is its original body position. In Jenkins’ (1970b: figs. 10 and 11) reconstruction, the ventral margin of the posterior process of the iliac blade in Massetognathus is nearly horizontal. This placement is probably reconstructed following the conditions represented in Thrinaxodon (AMNH 2228) and Diademodon (USNM 23352) where the pelvis is preserved in situ. Meanwhile, Bonaparte (1963: fig. 21) reconstructed the iliac blade as being more posteriorly inclined in Exaeretodon; Bonaparte (1966: fig. 15) and Kemp (1980: fig. 13) represented the ilium anteriorly inclined in Pascualgnathus and Luangwa, respectively. The exact original position of the ilium is difficult to infer, but the axis of the attaching points of the ribs on the iliac blade should form a small angle with the horizontal (Fig. 14).

Based on the reconstructed posture, the pubis extends anteriorly beyond the anterior margin of the acetabulum in Scalenodon, Andescynodon, and Pascualgnathus, and ventral to the acetabulum, without reaching its anterior margin, in Luangwa, Massetognathus, Menadon, and Exaeretodon. The pubis generally is ventrally and medially directed, but is almost medially directed in Luangwa. The diameter of the obturator foramen is similar to the diameter of the acetabulum in Scalenodon, Andescynodon, Pascualgnathus, and Massetognathus, smaller in Exaeretodon, and perhaps larger in Luangwa.

Hindlimb

As major supporting bones, the diameters of the femur, tibia, and fibula increase with length, so they are slender in small specimens and more massive in large ones (Table 3).

Table 3 Measurements of hindlimbs of traversodontids (in mm).

		FL	FP	FD	TL	TP	TD	FI	
Boreogomphodon	NCSM 20698	>20	10		22	5	4		
Pascualgnathus	MLP 65-VI-18-1	59	21	15	48	9.5	6.5	46	
Scalenodon	NHMUK R9391	85	29	22					
Luangwa	OUMNH TSK121	99	33	20					
Andescynodon	PVL 3894	42		11					
	PVL 3890	48	∼20	15					
Traversodon	GPIT RE 1069	122	55	40					
Massetognathus	PVL 5444	56		15					
Exaeretodon	PVL 2554	200	68	67	146	50	32	140	
	PVL 2162	172	61	52					
Notes.

FL femur length

FP femur proximal width

FD femur distal width

TL tibia length

TP tibia proximal width

TD femur distal width

FI fibula length

Femur

The basic structure of the femur in traversodontids is similar to that of Cynognathus (Fig. 15). The robust major trochanter generally is confluent with the head forming a semicircular outline to the proximal side of the bone. A notch separates the head from the trochanter major in Andescynodon and Massetognathus pascuali (Jenkins Jr, 1971: fig. 7) (Figs. 15D and 15F) but not in M. ochagaviae (Pavanatto et al., 2015: fig. 7). The notch could be the result of poor ossification or preservation, at least in Andescynodon. Kemp (1980) described the major trochanter of Luangwa as extending further proximally than in Cynognathus-Diademodon; however, the position of the major trochanter is similar between them and differs from most tranversodontids in being slightly more distal and lateral (Jenkins Jr, 1971: fig.48). The minor trochanter is mostly directed posteriorly and slightly medially in Boreogomphodon, Massetognathus, Pascualgnathus, Scalenodon angustifrons, and Traversodon, but is directed strongly medially in Andescynodon, Exaeretodon, and Luangwa. This morphology in the latter taxon could be accentuated by deformation.

Figure 15 Femora of traversodontids.

Pascualgnathus polanskii (MLP 65-VI-18-1), right side in (A) anterior and (B) posterior views; Scalenodon angustifrons (NHMUK R9391 = TR 8), left side in (C) anterior view; Luangwa drysdalli (OUMNH TSK121), left side in (D) anterior view; Andescynodon mendozensis (PVL 3894), left side in (E) anterior and (F) posterior views; Traversodon stahleckeri (GPIT RE 1069), right side in (G) anterior and (H) posterior views; Massetognathus pascuali (PVL no number), right side in (I) anterior and (J) posterior views; Exaeretodon argentinus (PVL 2554), left side in (K) anterior and (L) posterior views. All scale bars equal to 1 cm.

Pes

The tarsus is well preserved in Scalenodon (NHMUK R9391), Boreogomphodon (NCSM 20698), and Exaeretodon (PVL 2554). Seven tarsals are observed in the two former species, whereas one more is present in Exaeretodon (Bonaparte, 1963; Jenkins Jr, 1971). Metatarsal I is the shortest in all cases. The digital formula is interpreted as 2–3–3–3–3 for this group.

Conclusion

In summary, traversodontids share the following common postcranial features: 20–30 presacral vertebrae including seven cervicals; at least four sacrals; interclavicle cruciate with an elongate posterior ramus; scapula is distinctly constricted at the base of the acromion process, forming a neck; iliac dorsal margin nearly straight or slightly concave; major trochanter of femur robust; manus and pes digital formula 2-3-3-3-3. Variation is more extensive in the axial skeleton than in the pelvis and pectoral girdle. There are some important variations in the limbs. The vertebrae mainly differ in the number of sacral vertebrae, the presence of the anapophyses, and the angle of zygophyseal facets. The ribs in most species preserve the primitive morphology of Diademodon and trirachodontids while the ridge on costal plates is reduced. The structure of ribs is further reduced in some species like Boreogomphodon, Massetognathus, and Exaeretodon, but is complicated in Protuberum. The acromion process and the scapular neck are well developed in this group, but the relatively long neck only occurred in Boreogomphodon. The major transformation in shoulder girdle is the reduced size of the procoracoid. The anterior process of the iliac blade extends anteroventrally in this group, and the iliac neck is less pronounced than in the primitive member Pascualgnathus; the posterior process shows no distinct change other than the shortening in Exaeretodon. The structure of the limb bones is relatively uniform, and the robustness of the limb bones is directly related to their size.

The relative uniformity of the structures indicates similar locomotory strategies in this group. The humerus still moves in a horizontal plane, and the femur is half-erect. The anterior position of the iliac blade enables more efficient rotation of the femur in a nearly erect gait. The vertebral column is rigid but permits bending in most species, being more flexible in derived forms.

The cooperation and hospitality of the staff of various museums and institutions greatly facilitated our comparative studies. We would like to thank Tom Kemp (Oxford University Museum of Natural History, UK); the late Ray Symonds (University Museum of Zoology, Cambridge, UK); Sandra Chapman (Natural History Museum, London, UK); Fernando Abdala and Bruce Rubidge (Evolutionary Studies Institute, Johannesburg, South Africa); Jennifer Botha-Brink and Elize Butler (National Museum, Bloemfontein, South Africa); Roger Smith and Sheena Kaal (Iziko Museums–South African Museum, Cape Town, South Africa); Johann Neveling (Council for Geosciences, Pretoria, South Africa); Stephany Potze (Transvaal Museum, South Africa); Ana Maria Ribeiro (Museu de Ciências Naturais, Fundação Zoobotânica do Rio Grande do Sul, Porto Alegre, Brazil); Maria C. Malabarba (Museu de Ciências e Tecnologia, Pontifïcia Universidade Católica do Rio Grande do Sul, Porto Alegre, Brazil); Marina B. Soares and Cesar L. Schultz (Universidade Federal do Rio Grande do Sul, Porto Alegre RS, Brazil); Jaime E. Powell (Universidad Nacional de Tucumán, Argentina); Alejandro Kramarz and Agustín G. Martinelli (Museo Argentino de Ciencias Naturales “Bernardino Rivadavia”, Buenos Aires, Argentina); Guillermo F. Vega (Museo de Antropología, Universidad Nacional de La Rioja, Argentina); Ricardo Martinez (Museo de Ciencias Naturales, Universidad Nacional de San Juan, Argentina); Marcelo Reguero and Rosendo Pascual (Museo de La Plata, Argentina); Charles R. Schaff (Museum of Comparative Zoology, Harvard University, Cambridge, Massachusetts, USA), Hans-Dieter Sues and Matthew Carrano (National Museum of Natural History, Washington, D.C., USA), James A. Hopson (University of Chicago, Chicago, USA), Olivier Rieppel, Elaine Zeiger, and William F. Simpson (Field Museum of Natural History, Chicago, USA); and John Flynn (American Museum of Natural History, New York, USA). Fernando Abdala and Hans-Dieter Sues read the draft and gave helpful comments; Christian Kammerer and Leandro Gaetano reviewed the manuscript. Christian Kammerer kindly provided some photos. Gabe Bever improved the writing.

Institutional abbreviations

AMNH American Museum of Natural History, New York, NY, USA

BP Evolutionary Studies Institute, University of the Witwatersrand, Johannesburg, South Africa

GPIT Institut und Museum für Geologie und Paläontologie der Universität Tübingen, Tübingen, Germany

MACN Museo Argentino de Ciencias Naturales “Bernardino Rivadavia”, Buenos Aires, Argentina

NCSM North Carolina State Museum, Raleigh, NC, USA

NHMUK Natural History Museum, London, UK

NMQR National Museum, Bloemfontein, South Africa

PVL Colección de Palaeontología de Vertebrados, Instituto Miguel Lillo, Universidad Nacional de Tucumán, Argentina

USNM National Museum of Natural History, Washington D.C., USA

Additional Information and Declarations

Competing Interests

Author Contributions

Data Availability

The authors declare there are no competing interests.

Jun Liu conceived and designed the experiments, performed the experiments, analyzed the data, contributed reagents/materials/analysis tools, wrote the paper, prepared figures and/or tables, reviewed drafts of the paper.

Vincent P. Schneider performed the experiments.

Paul E. Olsen conceived and designed the experiments, performed the experiments.

The following information was supplied regarding data availability:

The raw data is included in the text and tables of the manuscript.

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
