# Peer review of "The postcranial skeleton of Boreogomphodon (Cynodontia: Traversodontidae) from the Upper Triassic of North Carolina, USA and the comparison with other traversodontids"

_PeerJ, doi:10.7717/peerj.3521_

## Round 0.1 · original submission · Major Revisions

Dear Dr Jun Liu

The Ms ID #15051 entitled "The postcranial skeleton of Boreogomphodon (Cynodontia: Traversodontidae) from the Upper Triassic of North Carolina, USA and the comparison with other traversodontids" which you submitted to PeerJ, has been reviewed by two reviewers and the Editor.

Both reviewers are coincident about the importance of the material described, particularly due to, in general, the scarcity of good descriptions of cynodont postcranial skeletons. Nevertheless, the Ms have two important general issues that need particular attention. One is the language, and as both reviewers strongly suggested, many sentences need to be rephrased in order to improve its readability. Please, pay particular attention to missspellings and grammar too and I encourage your co-authors (native English speakers) to help with this issue.

The second major issue are the figures, which are not very accurate and clear. They need to be formatted in a way that all the structures you discuss are perfectly clear in the pictures. To achive this important task you need to include good photographs, 1) avoiding blurry images (e.g. Fig 11.D and E), 2) too small pictures so the structures are very difficult to visualize (Fig. 1, Fig. 13), 3) avoid to make interpretative line drawings on the top of the picture (Figure 1) or to use color lettering very difficult to read (as the red letters in Figure 6), 4) the different pictures in a Figure need to be perfectly cropped, with clean edges and separated from each other (e.g. Figure 2 A and B, Figure 4E, Figure 5 A and D, Figure 7 B-D). You also need to standarized the figures so the formatting does not change so dramatically among the different Figures. Therefore, your must choose the type of scale bar (same high) with or without lettering (cm), and keep the same format and relative location in all the figures. Also, the capital letters used to identifiy each picture should be located in the same position in all figures. You need to construct your figures leaving space among the different pictures. Also, please as one of the reviewers pointed out, some abbreviations are not present in the epigraphs or are wrongly cited, and some taxa are misspelled in the epigraphs.

So, I am requesting that you revise carefully the parts mentioned above, simply to make your descriptions and conclusions clearer.

Thanks you for submitting your Ms to PeerJ and I look forward to receiving your revision.

Sincerely,
Dr. Claudia Marsicano

·

Basic reporting

This manuscript represents the long-awaited description of the postcrania of Boreogomphodon, a common cynodont from the Triassic of eastern North America whose rich fossil record is belied by the relatively few papers dealing with it. The description is adequately thorough, and covers all the elements represented for this taxon. Happily, the poorly-studied postcranial record of traversodontids in general is also given attention, with a long comparative section in the Discussion and figures of the diversity of these elements.

The discussion of postcranial diversity in traversodontids is unfortunately not well-served by the figures, which require major revisions to be useful and publishable. Several of the figures of the newly-described elements of Boreogomphodon are so blurry as to be useless (e.g., Fig. 11D) and new photographs are definitely needed. In other cases the elements are too small to be seen. Also, why are some figures of the new specimens in color, whereas others are in black and white (e.g., Fig. 5)? As a final note, I should say that the images of Menadon postcrania (from Kammerer et al., 2008) used in Figures 10 and 14 are still under copyright of the Society of Vertebrate Paleontology and cannot be reproduced without their express permission. I would, however, be happy to send you my original color photographs of those elements for use in revising the figure.

This manuscript requires heavy editing for language. I have noted some specific changes to be made in my line-by-line notes, but in many cases the phrasing of whole sentences needs detailed attention, and I would request that all three authors of the current manuscript go through it carefully for spelling, grammar, and flow.

Line 27: “Cynodontia is…a key component”—key component of what?
Lines 35: Rephrase this as “Traversodontids are the most successful Triassic cynodont group in terms of species richness and specimen abundance.”
Line 38: “described after”—change this to “described from” or “described based on”
Line 63: here, and elsewhere in the manuscript—the correct spelling of the species name for this taxon is cabralense (justified emendation because the genus name is neuter)
Line 74: “include” should be “includes”
Line 89: Could you include a table of important measurements for the specimens under study? You include a table on humeral ratios for traversodontids, but basic measurements just for the Boreogomphodon material would be extremely useful.
Line 94: Should change this to “only described”, as I have seen intact axial neural spines of Massetognathus in Argentina, and I believe Chris Sidor has an intact axial spine for one of his Tanzanian traversodontids.
Line 101: Again, a table of measurements would be very useful here.
Line 121: “presents” should be “present”
Line 131: add period after “vertebra”
Lines 131–132: This sentence doesn’t make sense—what is meant by “featured by”? Change to something like “they exhibit” or “the morphology of the vertebrae is generally characterized by”
Line 196: “tubercle” is misspelled
Line 201: “spatulated” should be “spatulate”
Line 217: change “The articulated hands” to “Articulated hands”
Line 230: “as well ossified” should be “are as well-ossified as in”
Line 253: “of ectepicondylar” should be “of the ectepicondylar”
Line 301: Table of measurements also would be helpful here.
Lines 317, 318: should be “Pelvic Girdle”
Line 349: here, and elsewhere, this should be “articulated with”
Line 353: “thicken” should be “thickened”
Line 369: as much as I enjoy this descriptor, you need a more “scientifically rigorous” way to describe this morphology (because it could be any kind of sausage and they curve in different ways—bratwurst? knockwurst? thüringer?)
Line 382: “ectocuneiform” is misspelled
Line 405: I think “event tough” was meant to be “even though”
Line 417: “traversodontid” is misspelled; also it is unclear what is meant by this line—are you talking about the hypothesized ancestral condition, or some kind of traversodontid bauplan?
Lines 432, 434, 439, 582, 587: “trirachodontid/s” is misspelled
Lines 438, 439: “plate” should be “plates”
Lines 447–448: either “ribs” should be “rib” or “has” should be “have”
Line 449: “has more expanded” should be “has a more expanded”
Line 450: “than remaining ones” should be “than the subsequent ones”
Line 470: “are common” should be “are a common”
Line 471: rephrase this whole sentence—start with the references, saying “Damiani et al. (2003)…demonstrated that Thrinaxodon inhabited burrows…etc.”
Line 479: “pattern is same” should be “pattern is the same”
Line 482: “inner organisms”, I think you mean “internal organs”
Line 502: “does no constrain below the acromion” should be “are not constricted below the acromion”
Line 506: “Kemp” is misspelled
Lines 517, 530: “possible” should be “possibly”
Line 536: “a flange” should be “of a flange”
Line 538: do you mean “better developed”?
Line 547: Where did you get this formula?
Line 568: delete extra spaces
Lines 572, 586: delete extra period
Line 572: should be “and Cricodon”
Line 580: “traversodontids” is misspelled
Lines 599, 601: “of iliac blade” should be “of the iliac blade”
Line 649: “constrains” should be “constricts”
Line 657: “Protuberum” is misspelled
Line 664: “uniform” should be “uniformity”
Line 671: “anteriorly” should be “previously”
Line 673: “have referred” should be “have been referred”
Line 694: should note here that Ray Symonds is sadly deceased

Experimental design

No comment.

Validity of the findings

No comment.

·

Basic reporting

In this manuscript, the authors describe the postcranial anatomy of a cynodont specimen tentatively assigned to the traversodontid Boreogomphodon in a comparative framework. In my opinion, this manuscript is very interesting and worth of publication due to the scarcity of published information regarding the usually neglected postcranial anatomy of cynodonts. Although the manuscript is in general terms well written, there are some issues that need to be addressed, particularly regarding the quality of the figures. Hence, my recommendation is that this manuscript is accepted after major changes are made. I have many comments and suggestions that are to be found in the Word file attached, the more important of them follow:
(1) I think that the manuscript would benefit from a fair amount of editing and re-wording. I have made some suggestions in this sense but the opinion of a native English speaker will substantially improve the manuscript readability.
(2) There are many statements which are not supported by the evidence or references provided.
(3) Proper justifications and discussion must accompany all statements, particularly in the “Conclusion” section. Some conclusions seem to be unrelated to the work performed. The authors must specifically discuss all the points present in the “Conclusion” section in a detailed manner in in the corresponding section of the manuscript and provide the adequate context for each of the conclusions.
(4) I strongly encourage the authors to include a table with the hindlimb measurements.
(5) Figure 1: the photograph provided is very difficult to understand. As it is, the bones are not easily identifiable. I strongly encourage the authors to also include a line drawing with their interpretation of the specimen.
(6) Figure 7: The features the authors mention in their description are not observed in the photograph and are not represented in the line drawing. I strongly encourage the authors to provide a close-up of the astragalus and a more detailed drawing with all the recognized structures labeled.
(7) There are several general problems with the figures/tables/epigraphs: (I) most of the photographs are out of focus, (II) the bones are not properly cropped, (III) there are dark and clear lines and spots surrounding the bones, (IV) the lettering (position and order) is not consistent among the different figures, (V) some labels are lacking a line pointing to the referred structure, (VI) some structures are not labeled and not properly visible due to the small size and/or lack of focus of the figured elements, (VII) some abbreviations are not present in the epigraphs or are wrongly cited, and (VIII) some genera are misspelled in the epigraphs. The authors should check carefully all the figures and tables for these and other problems.

My identity can be revealed to the authors. Please, do not hesitate in contacting me should any question arise.

Experimental design

no comment

Validity of the findings

no comment

Additional comments

no comment

---

## Round 0.2 · Minor Revisions

Dear Dr Jun Liu

The new version of your Ms has been reviewed by a reviewer and myself as Academic Editor. As a result only minor changes are necessary in the Ms.

Please read carefully in the annotated Ms the changes made by the reviewer.

Regarding to the figures, they still have some minor problems, which can be easily fixed as follows:

1) the scales are still not completely formatted in the same way in all the figures. I ask you to eliminate from figures 1 to 7 the number (e.g. 1 cm) on the top of the bars and reference the measurement in the corresponding figure legend thus will equal the same formatting as figures 8-11 and 13-15.
2) As you explain, you are not able to provide new better pictures of the humerus (Figure 5) in the near future. Nevertheless, you can try to make it better in order to improve the final result. Therefore, I ask you careful clean the border of each separate picture of the femur avoiding
a) to leave fragments of the original picture floating in the black background
b) to erase the real borders of the femur leaving a rugged contour. It is important that the picture display the actual material as it is.

Thanks you for submitting your Ms to PeerJ and I look forward to receive your new revision soon.

sincerely, Claudia Marsicano

·

Basic reporting

In this manuscript, the authors describe the postcranial anatomy of a cynodont specimen assigned to the traversodontid Boreogomphodon in a comparative framework. The scarcity of published information regarding the usually neglected postcranial anatomy of cynodonts makes this manuscript very interesting and worth of publication. This is the second time that I review this manuscript and the authors have performed most of the changes I suggested. In my opinion, only minor corrections (to be found in the annotated pdf file attached) are needed before publication.

Experimental design

no comment

Validity of the findings

no comment

Additional comments

no comment

---

## Round 0.3 · accepted · Accept

Dear Dr Jun,

It is a pleasure to accept your Ms # 15937, co-authored with V. Schenider and P. Olsen, entitled "The postcranial skeleton of Boreogomphodon (Cynodontia: Traversodontidae) from the Upper Triassic of North Carolina, USA and the comparison with other traversodontids" which you submitted to PeerJ.

Thank you for your fine contribution. We look forward to your future contributions to the Journal.

cheers, Claudia Marsicano